# Diagnostic Performance of Multispectral SWIR Transillumination and Reflectance Imaging for Caries Detection

**DOI:** 10.3390/diagnostics13172824

**Published:** 2023-08-31

**Authors:** Yihua Zhu, Chung Ng, Oanh Le, Yi-Ching Ho, Daniel Fried

**Affiliations:** 1Department of Preventive and Restorative Dental Sciences, University of California, 707 Parnassus Ave, San Francisco, CA 94143, USA; yihua.zhu@ucsf.edu (Y.Z.); chung.ng@ucsf.edu (C.N.); oanh.le@ucsf.edu (O.L.); yi-ching.ho@ucsf.edu (Y.-C.H.); 2Department of Stomatology, Taipei Veterans General Hospital, Taipei 11217, Taiwan

**Keywords:** SWIR imaging, caries detection, reflectance, transillumination

## Abstract

The aim of this clinical study was to compare the diagnostic performance of dual short wavelength infrared (SWIR) occlusal transillumination and reflectance multispectral imaging with conventional visual assessment and radiography for caries detection on premolars scheduled for extraction for orthodontics reasons. Polarized light microscopy (PLM) and micro-computed tomography (microCT) performed after tooth extraction were used as gold standards. The custom-fabricated imaging probe was 3D-printed and the imaging system employed a SWIR camera and fiber-optic light sources emitting light at 1300 nm for occlusal transillumination and 1600 nm for reflectance measurements. Teeth (n = 135) on 40 test subjects were imaged in vivo using the SWIR imaging prototype in the study and teeth were extracted after imaging. Our study demonstrates for the first time that near-simultaneous real-time transillumination and reflectance video can be successfully acquired for caries detection. Both SWIR imaging modalities had markedly higher sensitivity for lesions on proximal and occlusal surfaces compared to conventional methods (visual and radiographic). Reflectance imaging at 1600 nm had higher sensitivity and specificity than transillumination at 1300 nm. The combined SWIR methods yielded higher specificity but the combined sensitivity was lower than for each individual method.

## 1. Introduction

Despite the introduction of fluoridated drinking water, fluoride dentifrices and rinses, and improved dental hygiene in the US, dental decay continues to be the leading cause of tooth loss even though caries progression is potentially preventable and reversible if detected early [1,2]. The locations of most newly discovered caries are on the occlusal pits/fissures of the posterior dentition and the interproximal contact sites between adjacent teeth where physical access is difficult. Radiographs have low diagnostic sensitivity due to overlapping enamel [3,4]. Stains interfere with visual diagnosis and other proposed imaging technologies such as fiber optic transillumination or fluorescence-based methods and may increase false positives [5]. Likewise, early lesions on proximal surfaces can also be difficult to detect due to lack of physical access and the challenge of radiographically identifying the initial, subtle mineral loss of these lesions [3,4]. Radiographs are not effective for the early detection of occlusal caries lesions because of the overlapping features of the crowns [6,7,8]. Lesions are not typically visible on radiographs until decalcification has exceeded 30% [3,4], while SWIR imaging methods are diagnostic with only 5% decalcification [9]. Moreover, the risk of exposure to ionizing radiation is poorly understood, and even greatly reduced levels of radiation exposure may still pose a significant risk, especially for children and pregnant women. It is unlikely that improvements in radiographic sensitivity will enable the detection of hidden occlusal lesions or an improved diagnosis of suspicious occlusal caries lesions, those lesions with no cavitation and no radiographic radiolucency, but where caries is suspected due to surface roughness, opacities, or staining [6,7,8].

A highly sensitive and more reliable method for detecting demineralization during early lesion formation would be valuable for clinicians to monitor the activity of new caries and to assess the efficacy of non-surgical intervention. Short wavelength infrared (SWIR) and near-IR imaging (NIR) methods have been under development for almost 20 years for use in dentistry and several NIR clinical devices are now available commercially. Due to the high transparency of enamel at longer wavelengths, multiple imaging configurations are feasible. Lesions can be imaged using transillumination and reflectance from tooth occlusal, buccal, and lingual surfaces [9]. In proximal transillumination, the light source and detector are placed on the buccal and lingual sides of the tooth. The positions can be alternated to obtain images of each surface since this method will have greater sensitivity for those lesions located closer to the detector [9]. This is the same imaging geometry used to acquire bitewing radiographs. Interproximal lesions, the lesions located at the proximal contact points in between teeth, can be imaged via all three imaging geometries. Transillumination of the proximal contact points between teeth can also be accomplished via occlusal transillumination by directing SWIR light below the crown while imaging the occlusal surface [9,10] (see Figure 1). The latter approach that was used in this study is capable of imaging occlusal lesions as well with high contrast [9]. The first clinical SWIR/NIR study was carried out in 2010; it was demonstrated that interproximal lesions that appeared on radiographs could be detected in vivo using proximal and occlusal transillumination imaging at 1310 nm with similar sensitivity [10]. Currently, the only clinical systems that are available operate at shorter NIR wavelengths at 830 and 780 nm [11,12,13,14]. The use of shorter wavelength 830 nm NIR light was first investigated more than 20 years ago when it was found that the contrast between sound and demineralized enamel was higher at longer SWIR wavelengths [9]. Shorter wavelengths allow the use of less expensive silicon-based detectors. However, longer wavelength SWIR light has significant advantages, stains interfere significantly at wavelengths less than 1200 nm [5] and the contrast between sound and demineralized enamel is markedly higher at wavelengths beyond 1400 nm in reflectance measurements [9].

Studies over the past decade have indicated that transillumination performs best at 1300 nm where the transparency of enamel is highest while the contrast of lesions on tooth surfaces imaged using reflectance continues to increase with increasing wavelength and is highest at 1950 nm [9]. Several studies have investigated multispectral measurements using either one modality or combining transillumination and reflectance measurements. Zakian et al. [15,16] used multiple wavelengths of SWIR hyperspectral reflectance images to estimate the severity of occlusal lesions. Since multispectral SWIR reflectance and transillumination experiments have demonstrated that the tooth appears darker at wavelengths coincident with increased water absorption, multispectral images can be used to produce increased contrast between different tooth structures such as sound enamel and dentin, dental decay, and composite restorative materials [9,16]. Combining measurements from different SWIR imaging wavelengths and comparing them with concurrent measurements acquired by complementary imaging modalities should provide an improved assessment of lesion depth and severity. Radiographs markedly underestimate the depth and severity of interproximal lesions and clinicians assume that lesions penetrate much deeper than indicated in radiographs [17,18,19]. SWIR occlusal transillumination and reflectance have been combined into a single probe and tested in vitro [20]. Different illumination wavelengths have been optimized for each imaging mode, namely SWIR wavelengths greater than 1400 nm for reflectance and 1300 nm for transillumination. Simon et al. [21] built a benchtop simultaneous SWIR reflectance and transillumination system with tunable filters that ranged from 830–1700 nm and showed that the combined images have potential for the diagnosis of occlusal lesions and simulated cavitated and noncavitated interproximal lesions. The purpose of this study was to evaluate the diagnostic performance of a dual SWIR occlusal transillumination and reflectance device in vivo on teeth scheduled for extraction for orthodontic reasons. This is the first clinical study to employ dual SWIR transillumination and reflectance integrated into a single imaging device. In a prior SWIR clinical imaging study, separate occlusal and proximal imaging devices operating at 1300 nm were used along with reflectance imaging at 1450 nm [22]. That study showed that occlusal transillumination and reflectance imaging both had higher performance than proximal transillumination. In addition to their higher performance, it is much easier to combine occlusal transillumination and reflectance imaging into a single device. That study also showed that each device performed as well as radiographs for proximal lesions and if multiple devices were combined the sensitivity was much higher than for radiography [22]. Higher sensitivity may be of potential concern if it leads to over-treatment due to a high number of false positives. We hypothesize that the use of a single SWIR occlusal transillumination and reflectance imaging will increase the specificity and reduce the potential for false positives.

## 2. Materials and Methods

### 2.1. Participant Recruitment and Procedures

Study participants (n = 40 with 139 premolars) aged 12–60 with 2–4 premolars scheduled for extraction for orthodontic reasons were recruited from the UCSF Orthodontic Clinic by the study investigators (UCSF IRB 19-27656). One participant was lost to follow-up after the first visit and was excluded from the study. A total of 39 participants (n = 39 with 135 premolars) completed the study. Color images of each tooth were acquired using a FocusDent MD740 (Vilnius, Lithuania) 1280 × 960 pixel intraoral camera, and lesions were visually assessed. After extraction and before sectioning teeth were mounted in black Delrin blocks and imaged with digital radiographs using a CareStream 2200 System from Kodak (Rochester, NY, USA) operating at 60 kV. Radiographical contrast was calculated using (I_S_-I_L_)/I_S_ for each proximal contact. Where I_L_ is the lesion area and I_S_ is the sound control area. Areas for each I_L_ and I_S_ measurement were selected using PLM and microCT. I_L_ was measured as the mean intensity over the lesion area at each proximal contact and I_S_ was selected at a position either directly below or directly above I_L_ to ensure a similar enamel thickness. If no lesion was present the mean intensity was measured in an area of 20 × 20 pixels at the contact area to serve as I_L_ and an adjacent area was chosen for I_S_.

### 2.2. Design and Fabrication of the Dual Reflectance and Occlusal Transillumination SWIR Probe

The dual probe design consists of a reflectance probe body and an occlusal transillumination attachment, shown in Figure 1. The reflectance probe body and the occlusal transillumination attachment were 3D printed and are autoclavable. There is an air nozzle near the mirror to prevent fogging of the mirror. The air nozzle can also be used to dry the lesion to increase lesion contrast and potentially assess lesion activity.

The dual reflectance and occlusal transillumination SWIR probe was designed in Fusion 360 from Autodesk (San Francisco, CA, USA). The handpiece was fabricated using a Formlabs (Somerville, MA, USA) Form 3 Low Force Stereolithography 3D printer. The probe consists of two components, a main body containing the light source for reflectance and the light collection optics and a second attachment containing the transillumination light source. Details regarding the fabrication of the dual occlusal transillumination and reflectance imaging probe system and optical probe have been previously described [20].

### 2.3. Image Acquisition and Analysis

The SWIR images were captured using a 640 × 480 pixel micro-SWIR camera (SU640CSX) measuring only 32 × 32 × 28 mm from Sensors Unlimited (Princeton, NJ, USA). Two planoconvex antireflection coated lenses of 60 and 100 mm focal length along with an adjustable aperture were placed between the handpiece and the InGaAs camera to provide a field of view of 11 mm^2^ at the focus plane. A low-OH optical fiber of 1 mm diameter was used to deliver light from a 1604 nm superluminescent diode (SLD), Model ESL 1620-2111 from Exalos (Schlieren, Switzerland) with an output of 17 mW and a bandwidth of 46 nm. The intensity delivered to the tooth was 5 mW. The 1600 nm light passes through a polarizing beam-splitting cube before incidence on the tooth and a linear polarizer was placed before the camera to achieve cross-polarization for glare reduction. The transillumination light was delivered through two 0.4 mm diameter low-OH optical fibers. A 1314 nm (BW) SLD, Model DL-CS3452A-FP 1620-2111 from Denselight (Singapore) with an output of 48 mW and a bandwidth of 33 nm was used as the source for transillumination. A 50/50 beam splitter was used to deliver light to each arm for transillumination. The output intensity of each arm was set at 10 mw before entering the Teflon plugs located at the end of each arm.

Image processing of the images was performed by custom scripts written using MATLAB from Mathworks (Natick, MA, USA). The acquired 12 bit images (4096) were converted to 16 bit (65,535) by multiplying by 16 and subtracting 1 to facilitate processing using MATLAB. Contrast was calculated at 3 different lesion locations including occlusal grooves or fissures and the proximal mesial and distal contacts. The contrast was calculated for each location using the formula (I_L_ − I_S_)/I_L_ for reflectance images and (I_S_ − I_L_)/I_S_ for transillumination images. I_L_ was measured as the mean intensity over the lesion area. I_S_ was selected at adjacent confirmed sound locations [5]. I_L_ was also measured at the proximal contact or occlusal fissure even if no lesion was present in order to measure the contrast of sound areas for comparison. If no lesion was present the mean intensity was measured in an area of 20 × 20 pixels at the contact area to serve as I_L_ and an adjacent area was chosen for I_S_. Lesion areas were confirmed using the PLM and microCT images.

### 2.4. Sectioning, Polarized Light Microscopy (PLM) and Microcomputed Tomography (microCT)

The first 40 extracted teeth were sectioned and examined with polarized light microscopy (PLM). The remaining 95 teeth were examined intact using microcomputed X-ray tomography (microCT). The acquisition of an in-house microCT system mid-way through the study allowed us to switch to microCT. MicroCT does not require physical sectioning and avoids the risk of sample loss during sectioning. Samples (n = 40) were serially sectioned into ~200 μm thick mesiodistal slices using a linear precision saw, Isomet 5000 (Buehler, Lake Buff, IL, USA). PLM was used for histological examination of the thin sections using a Meiji Techno RZT microscope (Saitama, Japan) with an integrated digital camera, Canon EOS Digital Rebel XT (Tokyo, Japan). Sample sections were imbibed in deionized water and examined in the bright field mode with crossed-polarizers and a red I plate (550 nm retardation).

Whole teeth (n = 95) were imaged using microCT with a 10 μm resolution. A Scanco MicroCT 50 from Scanco USA (Wayne, PA, USA) was used to acquire the images. Acquisition parameters used for the microCT images were 90 kVP, 200 uA, 18 W, 20 FOV, 10 µm voxel size, 500 ms integration time, and an aluminum 0.5 mm filter.

## 3. Results

### 3.1. Imaged Teeth and Lesion Statistics

A total of 40 patients were recruited from the UCSF Orthodontics clinic. A total number of 135 teeth were imaged with 135 occlusal surfaces and 270 proximal surfaces for a total of 405 surfaces. However, 51 surfaces were excluded from the final analysis for the following reasons: pre-existing sealants, composite restorations, orthodontic bonding adhesive, tooth restored with composite after imaging, and tooth severely damaged during extraction. Hence, the total number of surfaces that were included in the final analysis totaled 354. The 354 surfaces were categorized based on surface type (sound tooth structure, enamel and dentinal lesions, and surface location (occlusal or proximal)). PLM/microCT identified 27 occlusal surfaces, 100 mesial proximal surfaces, and 86 distal proximal surfaces as sound tooth structures. Those surfaces are summarized in Table 1. There were 76 occlusal enamel lesions, 18 mesial enamel lesions, and 31 distal enamel lesions. Lesions penetrating the underlining dentin (dentinal lesions) were found on 11 occlusal surfaces, 3 mesial surfaces, and 2 distal surfaces. Cracks were found on 1 occlusal surface, 6 mesial surfaces, and 4 distal surfaces. One single distal cavitation was discovered by the microCT analysis that was also classified as one of the five proximal dentinal lesions. Visual and radiographic assessment was carried out by three clinical examiners using the intraoral color images and radiographs taken of the extracted teeth. The diagnostic performance of these conventional methods is summarised in Table 2.

Images from one tooth from the study with two interproximal lesions are shown in Figure 2. The color image in Figure 2A shows a bicuspid with a sealant on the occlusal surface along with a visible lesion on the distal surface. Neither lesion is obvious in the radiograph in Figure 2B. Only one of the three clinical examiners detected the lesion on the distal surface in the radiograph and by visual examination and none identified the lesion on the mesial surface by radiograph or visual examination. After tooth extraction and sectioning, PLM in Figure 2C shows that lesions are present on the distal and mesial surfaces. The SWIR reflectance and occlusal transillumination images acquired with the dual probe are shown in Figure 2D,E and both lesions are visible in the yellow circles.

Images from a second tooth from the study are shown in Figure 3. The color image in Figure 3A shows a bicuspid with hypomineralization and areas of high reflectivity near the mesial and distal surfaces indicated by the green arrows. No lesions are obvious in the radiograph in Figure 3B. Only one of the three clinical examiners detected the proximal lesion on the distal surface in the radiograph and none identified it by visual assessment. The SWIR occlusal transillumination and reflectance images acquired with the dual probe are shown in Figure 3C,D and both images show occlusal and proximal lesions in the positions of the yellow boxes. MicroCT images in Figure 3E,F show that lesions are present in the same areas of the yellow boxes. It is interesting that the bright areas in the color image of Figure 3A do not correspond to lesion areas. It is likely they are due to hypomineralization. The SWIR reflectance image in Figure 3C shows the triangular-shaped lesion at the distal contact matching the microCT image and shows no increased reflectivity at the mesial contact.

There were 16 lesions that penetrated into dentin, 11 occlusal lesions, and 5 lesions on proximal surfaces. None of the 11 occlusal dentinal lesions were detected by radiography. Radiography detected 3 out of 5 of the proximal lesions (mean of 3 examiners) and only 1 out of the 5 proximal lesions had a radiographic contrast greater than 0.1. Visual assessment (mean of 3 examiners) detected 1 out of 5 of the proximal lesions and 4 out of the 11 occlusal lesions. SWIR reflectance (contrast > 0.1) detected 6 of the 11 occlusal lesions and 5 out 5 of the proximal lesions while 8 of the 11 occlusal lesions and 3 out 5 of the proximal lesions were detected by SWIR occlusal transillumination (contrast > 0.1).

### 3.2. Diagnostic Performance Based on Contrast Thresholds

A total of 87 occlusal lesions (enamel and dentinal) were identified from PLM/MicroCT. Only one occlusal lesion was identified on radiographs by the three clinical examiners. The mean contrast of the occlusal lesions for SWIR reflectance was 0.18 ± 0.14 and that for occlusal transillumination was 0.16 ± 0.08. For both modalities, the contrast was significantly higher in lesion areas.

There were 54 proximal lesions that were detected using PLM/microCT and the mean ± (SD) of the radiographic contrast was 0.03 ± 0.04. The radiographic contrast for all the proximal lesions that were detected by at least one of the three clinical examiners (n = 23) was 0.07 ± 0.08 and the radiographic contrast for the proximal lesions that were detected by all three of the clinical examiners was (n = 3) 0.11 ± 0.06. The mean contrast for SWIR reflectance was 0.19 ± 0.15 and that for SWIR transillumination was 0.13 ± 0.08. These values are tabulated in Table 3.

Sensitivity, specificity, and accuracy were calculated for reflectance, transillumination, and both combined for varying contrast thresholds from 0.02 to 0.2. If the contrast was higher than that particular threshold it was considered a detected lesion. Plots of the accuracy versus contrast threshold are shown in Figure 4. An examination of Figure 4 shows that the accuracy reaches a plateau for lesions on proximal surfaces after a contrast threshold of 0.1 and drops after 0.1 for occlusal lesions on occlusal surfaces. Therefore, we chose a contrast threshold of 0.1 to calculate the diagnostic performance of reflectance, transillumination and both combined and the results are tabulated in Table 4.

## 4. Discussion

In this study, the diagnostic performance of a dual SWIR occlusal transillumination and reflectance imaging device was assessed in vivo on teeth scheduled for extraction based on orthodontic reasons. Caries status was unknown prior to imaging and PLM and microCT were used as gold standards. In this study, most of the 141 lesions were small and confined to the enamel. There were only 16 lesions that penetrated into the dentin and there was only one severe lesion that was cavitated and penetrated into the inner half of the dentin. The diagnostic performance of the SWIR methods was compared with conventional visual and radiographic methods. The motivation for combining reflectance and transillumination was to increase diagnostic performance by reducing false positives and providing more accurate measurements of lesion severity. The specificity did increase to 0.93 for proximal surfaces and 0.96 for occlusal surfaces for combined SWIR imaging. However, the accuracy for the combined methods was lower than for SWIR reflectance alone.

The posterior occlusal surfaces and interproximal surfaces between adjacent posterior teeth are challenging areas to detect early demineralization using radiography, due to the overlapping topography of the occlusal surface and the low sensitivity for detection of early proximal lesions that extend only into enamel. Visual detection of early demineralization on occlusal surfaces may also be confounded by staining in the occlusal fissures that make it difficult for clinicians to accurately pinpoint the areas of demineralization [5]. Stains also tend to make early lesions appear more severe. A thorough and complete removal of all stains from the pits and fissures is not clinically feasible. Visual and radiographic assessments were performed on the teeth by three clinical examiners. Based on the poor agreement of conventional diagnosis for which the examiners have had many years of experience, we decided to measure and compare the radiographic lesion contrast and the SWIR lesion contrast. In addition, in our prior dual transillumination and reflectance in vitro SWIR imaging study, there was poor agreement among clinical examiners which can be attributed to limited experience with new technology [20].

Prior studies have demonstrated that occlusal transillumination and reflectance imaging modalities using longer wavelengths may offer higher sensitivity regarding early lesion detection [11,12,14,22], but each individual imaging modality on its own has lower specificity than radiography and may lead to more false positives. We demonstrated in this study that it is possible to capture near-simultaneous occlusal transillumination (1300 nm) and reflectance (1600 nm) videos of lesions successfully in vivo, which has not been conducted prior to this study. Both imaging modalities, whether considered separately or when combined, had markedly higher sensitivity for lesions on both proximal and occlusal surfaces compared to conventional visual assessment and radiography. It is important to note that the teeth utilized in this study were subject to crowding and were extracted for orthodontic reasons. SWIR imaging was performed in vivo while radiography was performed on the teeth after extraction and did not suffer interference from crowding. Therefore, we suspect that the diagnostic performance of radiography would likely be even lower if it was performed in vivo with many overlapping teeth.

Higher sensitivity in detecting early lesions should be advantageous because clinicians can better assess the caries risk of the individual and intervene earlier with minimally invasive techniques to prevent such lesions from progressing further to avoid the need for surgical intervention. SWIR methods do not utilize ionizing radiation and are well suited for monitoring lesions over time to determine if they increase in severity.

The performance of reflectance imaging at 1600 nm was higher than for transillumination imaging at 1300 nm. One can argue that the performance of combined SWIR reflectance and transillumination was not any better than SWIR reflectance alone. Additional analysis needs to be performed to better assess the advantages of the combined imaging approach and fully evaluate the ability to assess lesion depth and severity. It was interesting that SWIR transillumination was most effective in detecting 8/11 of the deeper occlusal lesions that penetrated into the dentin.

SWIR imaging is particularly well suited for use with artificial intelligence (AI). Casalegno et al. showed that AI could be used to analyze clinical near-IR transillumination images at 780 nm [23]. The much higher lesion contrast of SWIR methods compared to radiography and the lack of interference due to stains are major advantages that can likely be exploited using AI approaches to identify lesions.

This study suggests that SWIR Imaging methods offer high sensitivity for lesions on proximal and occlusal surfaces without the interference of stains that appear at wavelengths of less than 1200 nm. No SWIR methods are currently available in the commercial market due to high cost and security concerns, but we anticipate that this innovative technology will increasingly become more accessible for medical use. Commercial NIR imaging devices for caries detection operate at 780 and 850 nm, where studies show that there is still significant interference from stains [5]. In addition, the contrast between sound and demineralized tooth structure has previously been shown to be markedly higher at longer SWIR wavelengths than it is at 780 or 850 nm [5,9]. The primary disadvantage of operating at longer SWIR wavelengths is that Si-based imaging technologies are only efficient at wavelengths under 1000 nm. Alternative imaging technologies such as InGaAs and Ge-enhanced Si are still expensive. The limited use of these more innovative semiconductor technologies is a major reason for the high cost. However, with expanded use, those prices are expected to decrease. The cost has decreased significantly in the past 10 years and the performance has increased markedly.

## 5. Conclusions

Combined SWIR occlusal transillumination and reflectance images yielded an increase in specificity for lesions on both occlusal and proximal surfaces. The sensitivity of the SWIR imaging methods was markedly higher than for conventional methods. The contrast of lesions in the occlusal pits and fissures and the proximal contacts where most lesions are located was significantly higher (*p* < 0.05) than sound areas for both SWIR occlusal transillumination and reflectance while there was no significant difference in contrast between lesion and sound areas for radiography.

## Figures and Tables

**Figure 1 diagnostics-13-02824-f001:**
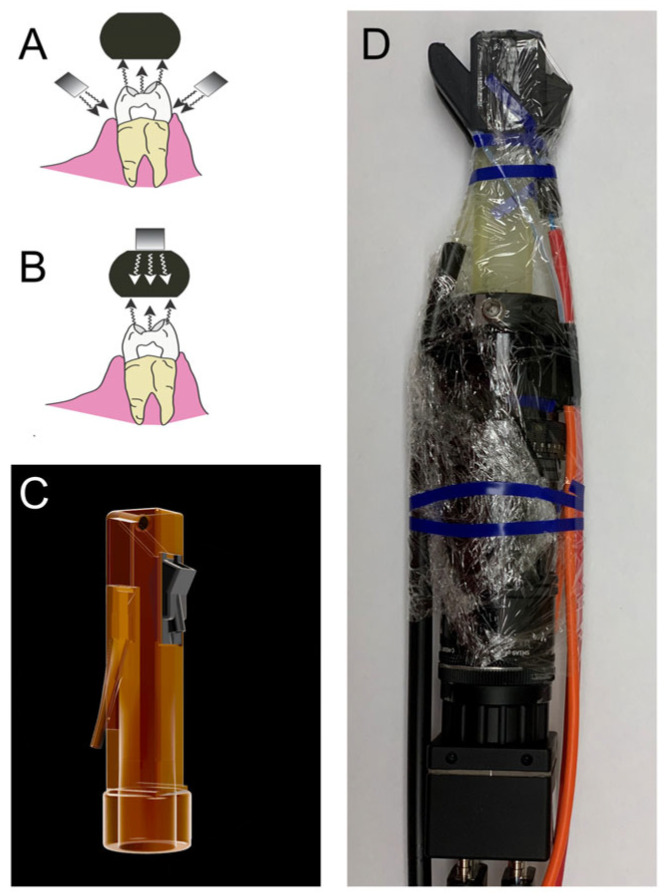
(**A**) Occlusal transillumination diagram, (**B**) reflectance diagram, (**C**) 3D printed handpiece attachment with insert (black) holding polarizing beam splitter cube. (**D**) Entire dual reflectance and transillumination wrapped for infection control and ready for clinical imaging.

**Figure 2 diagnostics-13-02824-f002:**
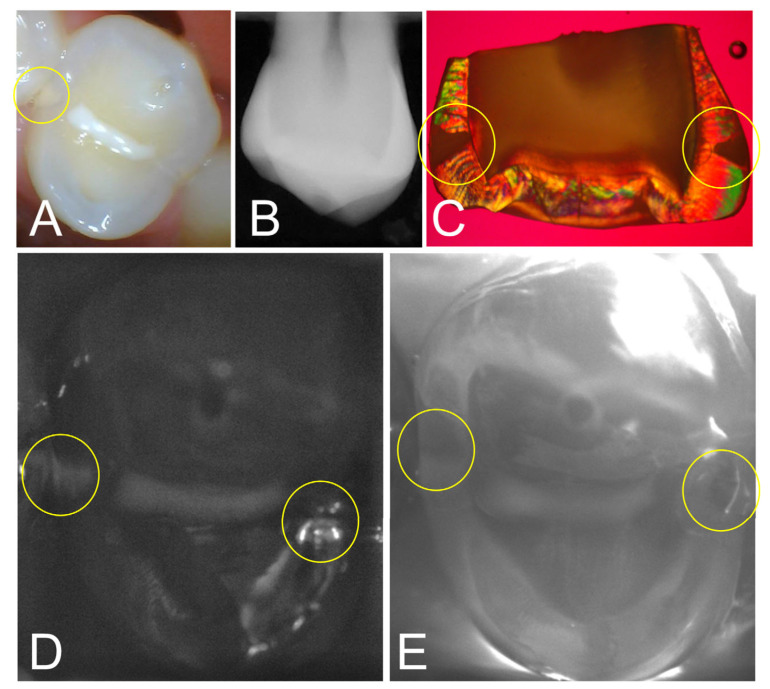
(**A**) Color image of a bicuspid with a sealant on the central fissure; a lesion is visible at the distal contact, while no lesion is visible at the mesial contact. (**B**) The radiograph shows no obvious lesions. (**C**) PLM image of a section cut from the tooth after extraction shows proximal lesions on both surfaces (yellow circles). (**D**) SWIR reflectance and (**E**) occlusal transillumination images show lesions on both surfaces.

**Figure 3 diagnostics-13-02824-f003:**
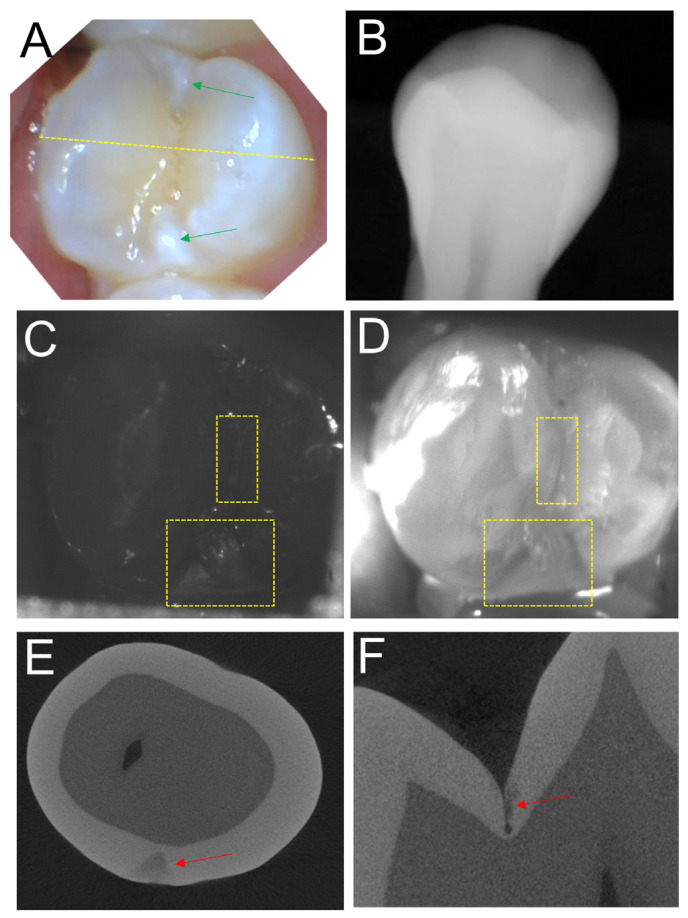
Images of a tooth with lesions on the occlusal and distal proximal surfaces The color image in (**A**) shows several bright areas of high reflectivity near the mesial and distal surfaces indicated by the green arrows that may be due to proximal lesions or hypomineralization. No lesions are visible in the radiograph in (**B**). SWIR reflectance (**C**) and occlusal transillumination (**D**) images show occlusal and proximal lesions in the positions of the yellow boxes. MicroCT images show lesions on the distal proximal surface (**E**) and the occlusal (**F**) surface as indicated by the red arrows.

**Figure 4 diagnostics-13-02824-f004:**
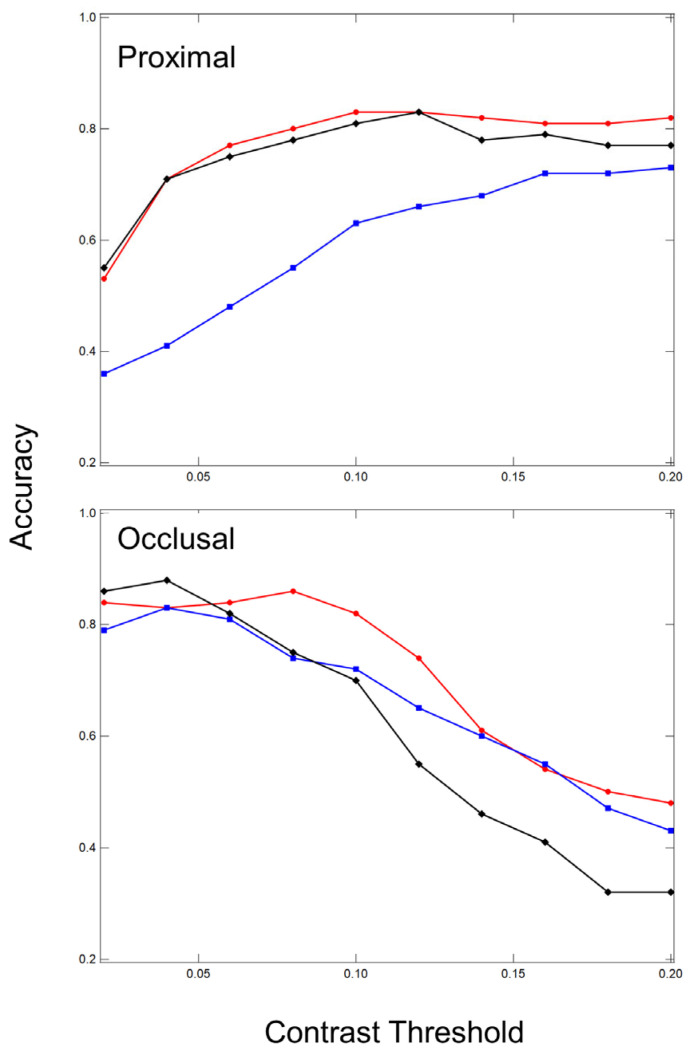
Plots of the accuracy calculated for SWIR reflectance (red), SWIR transillumination (blue), and the combined modalities (black) for the proximal (n = 54) occlusal lesions (n = 87) as a function of the lesion contrast threshold.

**Table 1 diagnostics-13-02824-t001:** Summary of proximal and occlusal surfaces based on PLM and microCT.

Surfaces	Occlusal	Proximal
**Sound**	27	186
**Enamel Lesions**	76	49
**Dentinal Lesions**	11	5
**Total**	114	240
**Rejected**	21	30

**Table 2 diagnostics-13-02824-t002:** Conventional visual and radiographic assessment of the lesions on the occlusal and proximal surfaces with three clinical examiners.

	Proximal (n = 54)	Occlusal (n = 87)
**Radiography**		
Mean Number of Detected Lesions	12	1
Accuracy	0.79	0.24
Sensitivity	0.22	0.01
Specificity	0.95	0.99
Interexaminer Reliability	0.93	0.99
**Visual Assessment**		
Mean Number of Detected Lesions	1	25
Accuracy	0.78	0.41
Sensitivity	0.02	0.29
Specificity	1	0.81
Interexaminer Reliability	1	0.85

**Table 3 diagnostics-13-02824-t003:** Mean lesion contrast ± s.d. for radiography and SWIR occlusal transillumination and reflectance at the three potential lesion locations with and without lesions present. For columns with an asterisk, the mean contrast was significantly higher for lesion areas (*p* < 0.05). For areas with no lesion present the contrast is the mean of the absolute value of the individual contrast values.

	Proximal			Occlusal	
	Radiography	Reflectance	Transillumination	Reflectance	Transillumination
**No Lesion**	0.021 ± 0.032	0.057 ± 0.075	0.10 ± 0.081	0.048 ± 0.045	0.088 ± 0.069
**Lesion**	0.030 ± 0.037	* 0.19 ± 0.15	* 0.13 ± 0.084	* 0.18 ± 0.14	* 0.16 ± 0.078

**Table 4 diagnostics-13-02824-t004:** Diagnostic performance of SWIR reflectance, occlusal transillumination, and combined methods along with radiography using a diagnostic contrast threshold of 0.10.

	Accuracy	Sensitivity	Specificity
** Proximal Surfaces **			
**Reflectance**	0.83	0.72	0.86
**Transillumination**	0.63	0.63	0.62
**Combined**	0.81	0.44	0.93
**Radiography**	0.79	0.08	0.99
** Occlusal Surfaces **			
**Reflectance**	0.82	0.80	0.89
**Transillumination**	0.72	0.75	0.63
**Combined**	0.70	0.62	0.96

## Data Availability

Not applicable.

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
