# Peer review of "Diagnostic Performance of Multispectral SWIR Transillumination and Reflectance Imaging for Caries Detection"

_diagnostics, 2023, doi:10.3390/diagnostics13172824_

Round 1

Reviewer 1 Report

The authors give a feasible new diagnostic method for the diagnosis of early caries. This method, if applied to the clinic, will bring the diagnosis of early caries into a completely new realm. However, the imperfection of the existing technology is also obvious.

The objective computational modeling of SWIR images in this study is a more desirable way of obtaining results compared to the authors' previous studies. However, the description of the SWIR contrast calculation lacks the necessary details.

1. There is a well-defined formula for radiograph contrast calculation in the study; does this formula serve equally well as a formula for SWIR contrast?

2. How do you identify sound /lesion areas on radiographs and SWIR images? How is the contrast calculated in Table 3? Calculate the average of 3 points in the lesion area? Please elaborate on the contrast calculation method.

3. Please explain that as contrast threshold increases, the diagnostic accuracy of occlusal surface lesions increases while the diagnostic accuracy of proximal surface lesions decreases? (Figure 4) The contrast of the lesion area in the image should represent the difference between the lesion area and the normal area. For proximal surface lesions, why is accuracy lower with higher contrast?

4. Two methods were used as gold standards in this study. How do you prove the consistency of the two gold standards? Micro-CT images are continuous images with 10 μm resolution. For PLM, how many slices were examined per tooth? Or all slices were examined?

5. Some minor typographical errors. Line 126, primary? Line 198, missing half a bracket. Line 209, what is TMR

Author Response

We have modified the manuscript as recommended and we are grateful for the reviewers’ helpful suggestions.  Text that has been added to the manuscript is highlighted in yellow in the amended manuscript.

Reviewer Comments:

Review 1

The authors give a feasible new diagnostic method for the diagnosis of early caries. This method, if applied to the clinic, will bring the diagnosis of early caries into a completely new realm. However, the imperfection of the existing technology is also obvious.

The objective computational modeling of SWIR images in this study is a more desirable way of obtaining results compared to the authors' previous studies. However, the description of the SWIR contrast calculation lacks the necessary details.

  1. There is a well-defined formula for radiograph contrast calculation in the study; does this formula serve equally well as a formula for SWIR contrast?

The contrast calculation equation for radiographs is the same as for SWIR transillumination, as the lesion is darker than sound for both modalities.  However, the lesion is brighter than sound for reflectance. (Lines 152-154) The contrast was calculated using the formula (IL − IS)/IL for reflectance images and (IS – IL)/IS for transillumination images, where IL is the mean intensity in the lesion area and IS is the mean intensity of the sound area.

  1. How do you identify sound /lesion areas on radiographs and SWIR images? How is the contrast calculated in Table 3? Calculate the average of 3 points in the lesion area? Please elaborate on the contrast calculation method.

Added to Section 2.1:

Areas for each  IL and IS measurement were selected using PLM and microCT.  IL was measured as the mean intensity over the lesion area at each proximal contact and IS was selected at a position either directly below or directly above IL to ensure a similar enamel thickness.  If no lesion was present the mean intensity was measured in an area of 20 x 20 pixels at the contact area to serve as IL and an adjacent area was chosen for IS.

Added to Section 2.3:

Contrast was calculated at 3 different lesion locations including occlusal grooves or fissures and the proximal mesial and distal contacts. The contrast was calculated for each location using the formula (IL − IS)/IL for reflectance images and (IS – IL)/IS for transillumination images.  IL was measured as the mean intensity over the lesion area.  IS was selected at adjacent confirmed sound locations [5].  IL was also measured at the proximal contact or occlusal fissure even if no lesion was present in order to measure the contrast of sound areas for comparison.  If no lesion was present the mean intensity was measured in an area of 20 x 20 pixels at the contact area to serve as IL and an adjacent area was chosen for IS.  Lesion areas were confirmed using the PLM and microCT images. 

Added to Table 3:

Mean lesion contrast ± s.d. for radiography and SWIR occlusal transillumination and reflectance at the three potential lesion locations with and without lesions present.  For columns with an asterisk the mean contrast was significantly higher for lesion areas (P< 0.05).   For areas with no lesion present the contrast is the mean of the absolute value of the individual contrast values.

Sound changed to no lesion in the first column of the table

  1. Please explain that as contrast threshold increases, the diagnostic accuracy of occlusal surface lesions increases while the diagnostic accuracy of proximal surface lesions decreases? (Figure

The labels in Figure 4 were switched and have been corrected.  The sensitivity falls off much faster with increasing contrast threshold for occlusal lesions.  Occlusal lesions are located at the occlusal surface while proximal lesions are located 1-3 mm below the occlusal surface.  The sensitivity decreases almost linearly for increasing contrast threshold from 0 to 0.2 for both occlusal and proximal lesions.  The rate of decrease is more gradual for proximal lesions since they are located well below the surface.   The specificity increases logarithmically increasing rapidly from 0 to 0.06 and slowly after that.

  1. The contrast of the lesion area in the image should represent the difference between the lesion area and the normal area. For proximal surface lesions, why is accuracy lower with higher contrast?

The labels in Figure 4 were switched and have been corrected.  The accuracy is lower for higher contrast for occlusal surfaces.  This is due to the very high initial sensitivity at low contrast since the lesions are located on the occlusal surface followed by the more rapid drop in sensitivity with higher contrast thresholds.

  1. Two methods were used as gold standards in this study. How do you prove the consistency of the two gold standards? Micro-CT images are continuous images with 10 μm resolution. For PLM, how many slices were examined per tooth? Or all slices were examined?

In this study we only report the lesion depth, i.e., whether it is confined to the enamel or penetrates to the dentin.  Both PLM and microCT are equally capable of providing lesion depth information.  The teeth were serial sectioned for PLM using an automatic saw yielding sections ~200 microns thick.   All intact sections were examined, 4-7 sections examined per tooth.  Some sections were lost due to fracture during cutting.  

  1. Some minor typographical errors. Line 126, primary? Line 198, missing half a bracket. Line 209, what is TMR?

The indicated errors have been corrected.  Primary molars has been replaced with premolars, bracket has been added and TMR has been removed from the table.  The amended manuscript has been proofed multiple times.

Reviewer 2 Report

Dear Authors,

This is a very interesting work, it is well described and very usful for clinicians. Early detection of caries and highly sensitive diagnostic methods are of clinical interest to the readers of the journal and fits the current trends in dentistry.

I only wanted to see a reference about the price of this device, because an interesting development could be to reduce the prices of these techniques.

Author Response

We have modified the manuscript as recommended and we are grateful for the reviewers’ helpful suggestions.  Text that has been added to the manuscript is highlighted in yellow in the amended manuscript.

Reviewer Comments:

Review 2

This is a very interesting work, it is well described and very useful for clinicians. Early detection of caries and highly sensitive diagnostic methods are of clinical interest to the readers of the journal and fits the current trends in dentistry.

I only wanted to see a reference about the price of this device, because an interesting development could be to reduce the prices of these techniques.

SWIR cameras of similar resolution now cost $10,000-20,000 but prices are decreasing rapidly as new technologies are being developed and there is increasing use and competition.  Here is a link to a recent overview of the technology and discussion on reducing costs.

New Sensor Materials and Designs Deepen SWIR Imaging Capabilities | Features | Jan 2023 | Photonics Spectra

https://www.photonics.com/Articles/New_Sensor_Materials_and_Designs_Deepen_SWIR/a68543#:~:text=This%20capability%20provides%20a%20useful%20metric%20for%20monitoring,and%20has%20inhibited%20wider%20use%20of%20the%20technology.